# Identifying Causal Effects
# via Context-specific Independence Relations

**Santtu Tikka**
Department of Mathematics and Statistics
University of Jyvaskyla, Finland
`santtu.tikka@jyu.fi`

**Antti Hyttinen**
HIIT, Department of Computer Science
University of Helsinki, Finland
`antti.hyttinen@helsinki.fi`

**Juha Karvanen**
Department of Mathematics and Statistics
University of Jyvaskyla, Finland
`juha.t.karvanen@jyu.fi`

## Abstract

Causal effect identification considers whether an interventional probability distribution can be uniquely determined from a passively observed distribution in a given causal structure. If the generating system induces context-specific independence (CSI) relations, the existing identification procedures and criteria based on do-calculus are inherently incomplete. We show that deciding causal effect non-identifiability is NP-hard in the presence of CSIs. Motivated by this, we design a calculus and an automated search procedure for identifying causal effects in the presence of CSIs. The approach is provably sound and it includes standard do-calculus as a special case. With the approach we can obtain identifying formulas that were unobtainable previously, and demonstrate that a small number of CSI-relations may be sufficient to turn a previously non-identifiable instance to identifiable.

## 1 Introduction

Statistical independence of random variables is a central concept in any data analysis and prediction task. An important generalization of this concept is context-specific independence (CSI) [26, 6]. For a simple example consider an antibiotic that normally has a dose–response effect on the number of bacteria. A genetic mutation makes the bacteria resistant to the antibiotic meaning that in the context of this mutation the dose and the number of bacteria are independent. CSI-relations have been utilized to analyze, for example, gene expression data [2], dynamics of pneumonia [33], prognosis of heart disease [22], proteins [15], parliament elections [22] and occurence of plants [22]. CSIs have also been used to speed up exact probabilistic inference [8, 12] and to improve structure learning [9, 19]. However, CSIs have received much less attention in causal inference and in particular, causal effect identifiability, despite their great potential in allowing for further identifiability results.

In the structural causal model (SCM) framework, the knowledge about causal mechanisms under investigation is represented as a directed acyclic graph (DAG). When some nodes represent unobserved latent variables, all information can be determined from a corresponding semi-Markovian graph. Assuming the qualitative information given by the graph, the aim in causal effect identification is to determine whether a causal effect $P(Y \mid \mathrm{do}(X), Z)$ can be uniquely determined from the available passively observed distribution. The known causal structure, whichever formalism is used, specifies (generalized) conditional independence properties of the system through d-separation. These warrant the manipulation of interventional distributions with the rules of do-calculus and thus the derivation

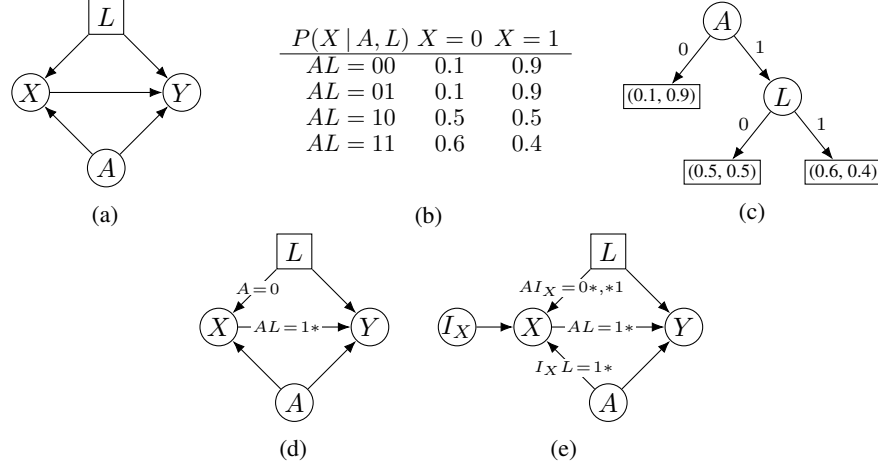

Figure 1: (a) L is latent unobserved variable. (b) CPT for $P(X \mid A, L)$. (c) Decision tree with $P(X \mid A, L)$ given in the leaf nodes. (d) corresponding labeled DAG (LDAG). (e) LDAG with an intervention node added for $X$.

of identifying formulas [23, 24]. The ID algorithm implements this inference: it can identify the causal effect whenever it can be non-parametrically identified [28, 17, 32].

When we have further information on the generative causal model, the completeness results of the previous approaches do not apply anymore: more causal effects become identifiable and do-calculus based methods will report false non-identifiability. One such piece of *still qualitative* information are CSI-relations. One example is shown in Fig. 1(a). The causal effect $P(Y \mid \text{do}(X))$ is non-identifiable by do-calculus here due to the back-door path through latent factor $L$. However, if we know CSIs $X \perp\!\!\!\perp L \mid A = 0$ and $X \perp\!\!\!\perp Y \mid A = 1, L$ the causal effect is identifiable (see Eq. 1 in Sec. 4.2).

Accounting for CSIs imposes additional challenges for deciding causal effect identifiability and to the derivation of identifying formulas. Instead of a graphical models for conditional independence, we need to employ inherently more complicated graphical models for CSI. As we shall show, derivation of causal effects requires context-specific reasoning. All this is well worthwhile if it warrants the identifiability of new causal effects.

We formulate the problem of causal effect identifiability in the presence of CSIs for binary variables and show that deciding non-identifiability is NP-hard (Sec. 3). Motivated by this we develop a calculus, and a search procedure over the rules of the calculus (Sec. 4 and 5). To make our search feasible, we eliminate redundant contexts, implement new separation criteria and use a well-motivated heuristic. With these techniques we scale up to network sizes often reported in literature. Most importantly, we show a host of examples where do-calculus cannot identify a causal effect but our search procedure leveraging on CSIs can prove identifiability (Sec. 6). Impact for future research and alternative approaches are discussed in Sec. 7.

## 2 Preliminaries: Graphical Models for Context-specific Independence

Our starting point is causal effect identification over a DAG $G = (\boldsymbol{V}, \boldsymbol{E})$. The set $\boldsymbol{W} \subseteq \boldsymbol{V}$ denotes a set of observed variables, marked by circular nodes. Since we also take into account the local structure, we mark any unobserved variables explicitly as rectangular nodes in the graph (as opposed to the semi-Markovian representation with bi-directed edges). The set $pa(Y)$ denotes the parents of a node $Y$ regardless of their observability. Notation $\boldsymbol{x}$ is used to denote an assignment to random variables $\boldsymbol{X}$, and $val(\boldsymbol{X})$ is used to denote the set of all possible assigments to $\boldsymbol{X}$. All variables are assumed to be binary.

There are different ways of representing the local structure in the local conditional probability distribution (CPD) of a node given its parents [20, 9]. One of the most popular ways of modeling the local structure is to cast some of the probabilities identical in the CPDs. For example the conditional

probability table (CPT) of $X$ in Fig. 1(b) has identical probabilities in the first two rows. One way to model such local structure is to use decision trees as in Fig. 1(c), see Koller et al. [20] for others.

Importantly, local structure induces local CSIs of the form $Y \perp\!\!\!\perp X \mid pa(Y) \setminus X = \boldsymbol{\ell}$, denoting that $Y$ is independent of the value of a parent $X$ when the other parents of $Y$ are assigned to values $\boldsymbol{\ell}$. The local CPT in Fig. 1(b) implies $X \perp\!\!\!\perp Y \mid A = 0$. The decision tree in Fig. 1(c) also shows this local CSI: once going down the branch with $A = 0$ the value of $X$ is not influenced by the value of $L$.

In this paper, we employ the idea of Pensar et. al. [25] and mark local CSIs as labels on the edges of the DAG. A DAG $(\boldsymbol{V}, \boldsymbol{E})$ together with a set of labels $\mathcal{L}$ defines a labeled DAG (LDAG) $G = (\boldsymbol{V}, \boldsymbol{E}, \mathcal{L})$, where for each edge $X \to Y \in \boldsymbol{E}$ there is a label $\boldsymbol{L} \in \mathcal{L}$, which is a (possibly empty) set of assignments to $pa(Y) \setminus X$ i.e., other parents of $Y$. Each assignment in the label encodes a local CSI: if $\boldsymbol{\ell} \in \boldsymbol{L}$, then $Y \perp\!\!\!\perp X \mid pa(Y) \setminus X = \boldsymbol{\ell}$. Symbol $*$ is used as a shortcut notation for any value. For example, the label $AL = 1*$ on $X \to Y$ in Fig. 1(d) implies that $X \perp\!\!\!\perp Y \mid A = 1, L$. Finally, throughout the paper, we restrict our attention to regular maximal LDAGs. Maximality requires that all labels that follow from other labels are recorded in the edges. Regularity means that edges absent in every context are not included in the graph. See Pensar et. al. [25] for details.

Any LDAG can be turned into a *context $\boldsymbol{s}$ specific DAG* by removing edges that are spurious (i.e., irrelevant) when variables $\boldsymbol{S}$ have values $\boldsymbol{s}$ as follows. The nodes appearing in the label $\boldsymbol{L}$ on some $X \to Y$ can be partitioned into two sets $\boldsymbol{A}$ and $\boldsymbol{B}$: nodes in $\boldsymbol{A}$ are assigned to $\boldsymbol{a}$ by the context $\boldsymbol{s}$, while nodes in $\boldsymbol{B}$ are not. Then, the edge $X \to Y \in \boldsymbol{E}$ is not present in the context $\boldsymbol{s}$ specific DAG (i.e., the edge is spurious) if $(\boldsymbol{a}, \boldsymbol{b}) \in \boldsymbol{L}$ for all possible assignments $\boldsymbol{b}$. For example, the context $A = 1$ specific DAG of Fig. 1(d) is identical to the underlying DAG except for $X \to Y$ being absent.

A *sufficient* condition for a non-local CSI to be implied by an LDAG structure is given by CSI-separation criterion [6]: If sets of nodes $\boldsymbol{X}$ and $\boldsymbol{Y}$ are d-separated given $\boldsymbol{C}, \boldsymbol{S}$ in the context $\boldsymbol{s}$ specific DAG of $G$, then $\boldsymbol{X} \perp\!\!\!\perp \boldsymbol{Y} \mid \boldsymbol{C}, \boldsymbol{s}$ is implied by $G$. Note that d-separation is a special case when $\boldsymbol{S} = \emptyset$. For example, the labeling in Fig. 1(d) implies that $X \perp\!\!\!\perp L \mid A = 0$ by this criterion, as the edge $L \to X$ is absent in the context $A = 0$ specific DAG.

We assume a positive distribution over the variables $\boldsymbol{V}$ [17]. This makes causal effects well-defined and justifies conditioning on any subset of variables or their particular assignments.

## 3 Causal Effect Identification for CSI-based Graphical Models

As the first contribution we formalize causal identifiability problem in the presence of CSIs. Identifiability [24, 29] considers whether a causal effect can be uniquely identified in models with a given fixed structure. If an effect is non-identifiable, there are (at least) two models that agree with the observations and have the same given structure but disagree on the causal effect.

We use LDAGs to define identifiability in the presence of CSIs, as LDAGs offer a simple and intuitive visual view of the causal structure and local CSIs. The LDAG is assumed known based on the background knowledge on the examined study, similarly as semi-Markovian graphs are standardly drawn for do-calculus. For example, consider (again) the case where an antibiotic $A$ had a dose-response effect to $H$ only if a genetic mutation $M$ had not taken place. Hence, we would mark label $M = 1$ on the edge $A \to H$. Thus, the causal effect identification problem can be formulated as:

**Input:** An LDAG $G$ over $\boldsymbol{V}$, $P(\boldsymbol{W})$ for $\boldsymbol{W} \subseteq \boldsymbol{V}$, a query $P(\boldsymbol{Y} \mid \mathrm{do}(\boldsymbol{X}), \boldsymbol{Z})$ s.t. $\boldsymbol{Y}, \boldsymbol{X}, \boldsymbol{Z} \subset \boldsymbol{W}$.

**Task:** Output a formula for $P(\boldsymbol{Y} \mid \mathrm{do}(\boldsymbol{X}), \boldsymbol{Z})$ over $P(\boldsymbol{W})$, or decide that it is non-identifiable.

When no labels appear on the edges of an LDAG, the causal structure can be directly cast as a semi-Markovian graph. Thus, the setting of do-calculus is a special case of this one.

### 3.1 On Computational Complexity

In contrast to causal effect identifiability over semi-Markovian graphs, which has polynomial decision procedures [28, 17], taking local structure and CSIs into account makes the corresponding decision problem NP-hard. (The proofs for all theorems are given in the supplementary material.)

**Theorem 1.** *Deciding non-identifiability of a causal effect given an LDAG over $\boldsymbol{V}$ and a passively observed distribution over $\boldsymbol{W} \subseteq \boldsymbol{V}$ is NP-hard.*

> **Rule 1** (Insertion/Deletion of observations):
> $$P(\boldsymbol{Y} \mid \mathrm{do}(\boldsymbol{X}), \boldsymbol{Z}, \boldsymbol{W}) = P(\boldsymbol{Y} \mid \mathrm{do}(\boldsymbol{X}), \boldsymbol{W}) \text{ if } \boldsymbol{Y} \perp\!\!\!\perp \boldsymbol{Z} \mid \boldsymbol{X}, \boldsymbol{W} \,\|\, \boldsymbol{X}$$
> **Rule 2** (Action/Observation exchange):
> $$P(\boldsymbol{Y} \mid \mathrm{do}(\boldsymbol{X}), \mathrm{do}(\boldsymbol{Z}), \boldsymbol{W}) = P(\boldsymbol{Y} \mid \mathrm{do}(\boldsymbol{X}), \boldsymbol{Z}, \boldsymbol{W}) \text{ if } \boldsymbol{Y} \perp\!\!\!\perp \boldsymbol{I_Z} \mid \boldsymbol{X}, \boldsymbol{Z}, \boldsymbol{W} \,\|\, \boldsymbol{X}$$
> **Rule 3** (Insertion/Deletion of actions):
> $$P(\boldsymbol{Y} \mid \mathrm{do}(\boldsymbol{X}), \mathrm{do}(\boldsymbol{Z}), \boldsymbol{W}) = P(\boldsymbol{Y} \mid \mathrm{do}(\boldsymbol{X}), \boldsymbol{W}) \text{ if } \boldsymbol{Y} \perp\!\!\!\perp \boldsymbol{I_Z} \mid \boldsymbol{X}, \boldsymbol{W} \,\|\, \boldsymbol{X}$$

Figure 2: Rules of do-calculus. The sets $\boldsymbol{X}, \boldsymbol{Y}, \boldsymbol{Z}$ and $\boldsymbol{W}$ are disjoint. Notation $\|\, \boldsymbol{X}$ means that the condition is evaluated in a graph in which edges into $\boldsymbol{X}$ are removed. $\boldsymbol{I_Z}$ denotes the intervention nodes of variables $\boldsymbol{Z}$ (see Sec. 4.1).

The proof of Theorem 1 shows that 3-SAT can be reduced to the identifiability of $P(Y \mid \mathrm{do}(X))$ from $P(X, Y)$. On an intuitive level, the intricate structure in the local CPDs allows for representing instances of NP-hard decision problems. This result is related to NP-hardness results of exact inference [10], implication problem of CSIs [20, 11] and the complexity results for Halpern's actual causation [1], however, we are not aware of other NP-hardness results for causal effect identifiability.

## 4 A Calculus for Determining Identifiability

In light of Theorem 1, fast algorithms for determining identifiability of a causal effects may be generally unobtainable. Thus, we take here an approach similar to [14, 23, 16] and formulate a calculus called CSI-calculus which can be used to show identifiability for particular instantiations of the problem. CSI-calculus is an extension of do-calculus of Fig. 2. In the first subsection we show that due to the versatile graphical model used (LDAG), we only need to consider identification of conditional probabilities (i.e., the do-operation is not needed). The second subsection gives the rules of CSI-calculus.

### 4.1 Reduction to the Identifiability of Conditional Probabilities in LDAGs

Interventions can be encoded naturally with the use of intervention variables and CSIs [6, 23, 13]. Here we show how this can be done for LDAGs.

For any LDAG $(\boldsymbol{V}, \boldsymbol{E}, \mathcal{L})$, we can construct an augmented LDAG that has the capacity to represent interventions as follows. Each node $X \in \boldsymbol{V}$ is augmented by an intervention node $I_X$ and an edge $I_X \to X$. If $I_X = 0$, then $X$ is in its passive observational state determined by its parents $pa(X)$. If $I_X = 1$, then $X$ is intervened on and its value is determined independently from its parents.

For every $X \in \boldsymbol{V}$ and every label $\boldsymbol{L_Z} \in \mathcal{L}$ of every incoming edge $Z \to X$ such that $Z \neq I_X$, we construct the augmented label $\boldsymbol{L}'_Z$ by including the assignments $I_X = *, pa(X) \setminus (I_X \cup Z) = \boldsymbol{\ell}$ for every $\boldsymbol{\ell} \in \boldsymbol{L_Z}$ and $I_X = 1, pa(X) \setminus (I_X \cup Z) = *$. In other words, $\boldsymbol{L}'_Z$ renders the edge $Z \to X$ spurious when $I_X = 1$ or in any context where $\boldsymbol{L_Z}$ would. Fig. 1(e) shows an LDAG that is constructed from the LDAG in Fig. 1(d) by adding an intervention node for $X$.

Using the above construction, an interventional distribution $P(Y \mid \mathrm{do}(X))$ is now simply a conditional distribution $P(Y \mid X, I_X = 1)$. Thus, we can essentially drop the do-operator from the problem definition, and model interventions using intervention nodes and CSIs instead. To simplify the notation, we omit intervention nodes for variables that are in their passive observational state from formulas. We do still include the do-operator when possible for improved readability.

### 4.2 Rules of the Calculus

Figure 3 describes the rules of CSI-calculus. In the rules we use terms that apply for all assignments (large letters) and to particular assignments (small letters). We do this in order to make the derivations shorter and identifying formulas more understandable. A valid calculus can be formed by omitting all large letters, but our experiments (Sec. 6) suggest that such a calculus is far less efficient.

Figure 3: Rules of CSI-calculus. The sets $\boldsymbol{X}_1, \boldsymbol{X}_2, \boldsymbol{Y}_1, \boldsymbol{Y}_2, \boldsymbol{Z}_1$ and $\boldsymbol{Z}_2$ are disjoint. We write $\boldsymbol{w}$ as shorthand for the explicit assignment $\boldsymbol{W} = \boldsymbol{w}$.

Rule 1 is directly the definition of context-specific independence which includes conditional independence as a special case. Rule 1 can be applied in both directions, when the term on the left is identified, so is the term on the right and vice versa, provided that the separation condition is satisfied.

Marginalization, conditioning and factorization from standard probability calculus are operationalized by rules 2–4, respectively. Rule 5 uses the law of total probability to obtain the probability of the complement. Rules 2–5 are applied from right to left: when the expressions on the right are identified, then so is the term on the left. Rule 5 is also valid when $\boldsymbol{Y}_1$ and $\boldsymbol{Y}_2$ are empty sets: in this case the rule should be understood as $P(1 - z \mid \boldsymbol{X}_1, \boldsymbol{x}_2) = 1 - P(z \mid \boldsymbol{X}_1, \boldsymbol{x}_2)$.

Rule 6 explicates that if we know the expression for each assignment $Z = z$ then we also know the expression without a specific assignment to $Z$. When rules 4–6 are applied, both distributions on the right-hand side must be known. Rules 7 and 8 formulate the fact that if an expression is known for all assignments to $Z$, it is also known for a specific assignment $Z = z$. For rules 5–8, it is assumed that $Z$ is a singleton for convenience. This assumption does not restrict identifiability since operations involving sets can be carried out by applying the rules for each member of the set sequentially. For identifiable queries, the formula in terms of the joint distribution $P(\boldsymbol{W})$ is easily obtained by backtracking the chain of manipulations that resulted in identification.

Importantly, CSI-calculus includes standard do-calculus of Fig. 2 as a special case.

**Theorem 2.** *CSI-calculus subsumes do-calculus.*

This means that any formula that is derivable with standard do-calculus over a DAG $G$ (w. latents), is also derivable using CSI-calculus over the LDAG formed by simply adding intervention nodes and labels as described in Section 4.1. After this augmentation, Rule 1 fully encompasses the three rules of do-calculus [23, 24]; this is shown in the proof of the theorem.

More importantly, the calculus of Fig. 3 can identify causal effects that are not identifiable with the standard do-calculus. For the example of Fig. 1, the following formula can be obtained:

$$P(Y \mid \operatorname{do}(X)) = P(Y \mid A = 0, X)P(A = 0) + P(Y \mid A = 1)P(A = 1). \tag{1}$$

A simple derivation of this formula using CSI-calculus is shown in Fig. 4. Note that the back-door formula $P(Y \mid \operatorname{do}(X)) = \sum_A P(A)P(Y \mid A, X)$ is not valid here: conditioning on X when $A = 1$ biases Y through $X \leftarrow L \rightarrow Y$.

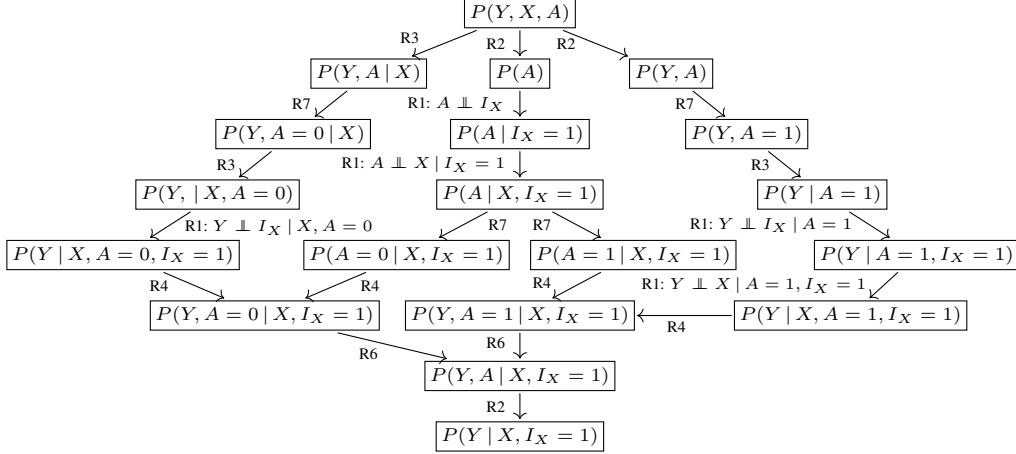

Figure 4: A derivation of $P(Y \mid \mathrm{do}(X))$ from $P(X, Y, A)$ in the example of Fig. 1. The applied rules and CSIs are marked next to the edges connecting the terms. The identifying formula is Eq. 1.

## 5 A Search for Causal Effect Identification

In contrast to the setting of standard do-calculus, due the formidable number of contexts and the causal structure being described by arguably more complex graph formalism, applying the rules of CSI-calculus by hand is impossible (recall also Theorem 1 on NP-hardness). Hence, we follow the approach of [30, 18] and devise a forward search procedure over the rules of CSI-calculus that is able to automatically output identifying formulas and derivations such as Fig 4.

However, for any instance, there are a vast number of terms that may end up being useful in identifying the query term; in fact, the derivation in Fig. 4 only shows the terms that were actually needed (in hindsight). For applying rule 1 we need to check a coNP-hard separation criterion, in contrast to the polynomial check of d-separation in the standard do-calculus setting. Hence, we focus here on how to efficiently evaluate separation criteria (Sec. 5.1), combine contexts (Sec. 5.2) and implement the heuristic search (Sec. 5.3) without weakening the theoretical properties (Sec. 5.4).

### 5.1 Implementing Separation Criteria

Rule 1 requires the evaluation of possibly non-local CSIs. Recall from Section 2, that CSI-separation is only a sufficient criterion; in practice it misses many of the important independence relations. For a feasible search procedure we need a sufficiently fast way to check a sufficient separation criterion. The following sufficient criterion is implemented in the search for this purpose.

**Theorem 3.** *If there exists a set $C$ such that $Y \perp\!\!\!\perp Z \mid X, w, C$ is implied by an LDAG $G$ and one of the following is also implied by $G$: (i) $Y \perp\!\!\!\perp C \mid X, w$, (ii) $C \perp\!\!\!\perp Z \mid X, w$, (iii) $Y \perp\!\!\!\perp C \mid X, Z, w$, or (iv) $Z \perp\!\!\!\perp C \mid X, Y, w$, then also $Y \perp\!\!\!\perp Z \mid X, w$ is implied by $G$.*

When a CSI statement $Y \perp\!\!\!\perp Z \mid X, w$ is encountered by the search, the following procedure is applied: First, we verify whether the CSI is directly encoded in a label. If it is, we can stop and if it is not, we continue by applying the CSI-separation criterion. If the CSI-separation criterion does not hold, we continue by attempting to find a set $C$ that satisfies $Y \perp\!\!\!\perp Z \mid X, w, c$ for all $c \in \mathrm{val}(C)$. Theorem 3 is then applied recursively to verify whether all of the required CSIs $Y \perp\!\!\!\perp C \mid X, w$, $C \perp\!\!\!\perp Z \mid X, w$, $Y \perp\!\!\!\perp C \mid X, Z, w$ or $Z \perp\!\!\!\perp C \mid X, Y, w$ hold in $G$. To guarantee that the recursion terminates, each variable can appear only once in each branch of the recursion. We further reduce the number of evaluated CSIs by caching them during the search.

### 5.2 Eliminating Redundant Contexts

The number of possible contexts increases exponentially with the number of variables. It is therefore important to determine which contexts should be considered when CSIs are evaluated. Different contexts often share the same context-specific DAG. We define the equivalence relation $\overset{s}{\sim}$ as follows:

**Algorithm 1**

---

**Input:** Target $Q = P(\boldsymbol{Y} \,|\, \mathrm{do}(\boldsymbol{X}), \boldsymbol{Z})$, LDAG $G$ and input $\boldsymbol{I} = \{P(\boldsymbol{W})\}$.
**Output:** A formula $F$ for $Q$ in terms of $P(\boldsymbol{W})$ or NA.
 1: **let** $\boldsymbol{U}$ be the set of unexpanded terms, initially $\boldsymbol{U} := \boldsymbol{I}$.
 2: **for** $P' \in \boldsymbol{U}$:
 3:     **let** $\boldsymbol{I}^*$ be the set of all distributions derived from $P'$ using the rules of Section 5.
 4:     **for** each new candidate distribution $P^* \in \boldsymbol{I}^*$, **do**
 5:         **if** an additional input is required that is not in $\boldsymbol{I}$, **then continue**.
 6:         **if** CSI relation of the current rule is not satisfied by $G$, **then continue**.
 7:         **if** $P^* = Q$, **then** derive a formula $F$ for $Q$ by backtracking and **return** $F$.
 8:         Add $P^*$ to $\boldsymbol{I}$, add $P^*$ to $\boldsymbol{U}$.
 9:     Mark $P'$ as expanded: remove $P'$ from $\boldsymbol{U}$.
10: **return** NA.

---

$\boldsymbol{s}_1 \overset{s}{\sim} \boldsymbol{s}_2$ if and only if the context $\boldsymbol{s}_1$ specific DAG is the same as the context $\boldsymbol{s}_2$ specific DAG, where $\boldsymbol{s}_1, \boldsymbol{s}_2 \in val(\boldsymbol{S})$. When evaluating the CSI $\boldsymbol{Y} \perp\!\!\!\perp \boldsymbol{Z} \,|\, \boldsymbol{X}, \boldsymbol{w}, \boldsymbol{C}$ of Theorem 3, we do not have to determine d-separation for every $\boldsymbol{c} \in val(\boldsymbol{C})$ and $\boldsymbol{w}, \boldsymbol{c}$ specific DAG. It suffices to restrict our attention to the context-specific DAGs given by the representatives of $val(\boldsymbol{C})/\overset{s}{\sim}$.

**Theorem 4.** *Let $\boldsymbol{R}$ be a set of representatives of $val(\boldsymbol{C})/\overset{s}{\sim}$. If $\boldsymbol{Y}$ is CSI-separated from $\boldsymbol{Z}$ by $\boldsymbol{X}$ in the context $\boldsymbol{w}, \boldsymbol{c}$ in $G$ for all $\boldsymbol{c} \in \boldsymbol{R}$, then $\boldsymbol{Y}$ is CSI-separated from $\boldsymbol{Z}$ by $\boldsymbol{X}$ in the context $\boldsymbol{w}, \boldsymbol{c}$ in $G$ for all $\boldsymbol{c} \in val(\boldsymbol{C})$.*

The definition of intervention nodes can also be used in this way. In general, an arbitrary context $\boldsymbol{S} = \boldsymbol{s}$ can render a number of edges spurious in the LDAG. However, if the context contains the assignment $I_X = 1$ for any node $X$, we know that every incoming edge of $X$ except $I_X \to X$ will be made spurious by definition without requiring any further verification.

## 5.3 Implementing the Search

Algorithm 1 shows the pseudo-code which implements the calculus of Section 4 and is capable of solving problems that fall under the formulation of Section 3 through the use of a search heuristic and elimination of redundant contexts. A single distribution is called a *term*, which is considered *expanded* if every valid manipulation has been performed on it.

The input distribution is marked as unexpanded on line 1 and iteration over the unexpanded terms begins on line 2. In order to guide the search to identify the most promising terms, we relate the identified distributions to the target $Q$ through a heuristic proximity function and always expand the closest term in $\boldsymbol{U}$ first. Note that if we were to expand *only* the closest term to the target greedily, several identifiable instances would be left non-identified because the identifying formulas and derivations are highly non-trivial. More details about the proximity function are given in the supplementary material. If multiple terms share the maximal value of the proximity function, the term that was identified first is selected. Next, the rules of Section 4 are applied to $P'$ and the derived candidate distributions are added to the set $\boldsymbol{I}^*$ on line 3. Note that not every distribution in $\boldsymbol{I}^*$ is necessarily identified at this point.

Iteration over the set $\boldsymbol{I}^*$ begins on line 4. Here the candidate terms $P^*$ in $\boldsymbol{I}^*$ that can be identified are added to the set $\boldsymbol{I}$. Previously identified terms are not identified again. Line 5 verifies that both required terms are identified for rules 4–6. Line 6 applies Theorem 3 to check the required CSI relation for rule 1. Tests for d-separation are carried out via relevant path separation [7].

If all requirements are met, $P^*$ is identified either as the target on line 7 or as a new unexpanded distribution on line 8. Once all candidate distributions are processed, we mark $P'$ as expanded on line 9. Note that $P'$ can still appear as a second required term on line 5 when another term is being expanded. Finally, if the target was not identified and the set of unexpanded distributions was exhausted, we deem the target non-identifiable by the search and return NA on line 10.

## 5.4 Theoretical Properties

The formulated search is sound in the following sense.

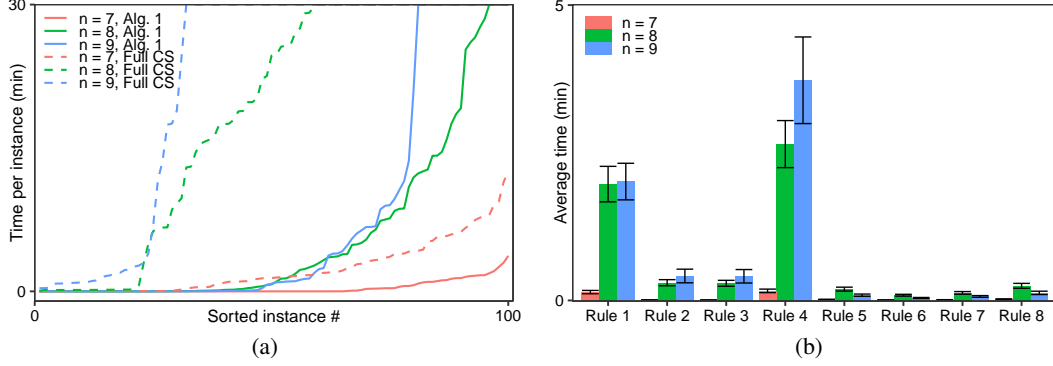

Figure 5: (a) Running times of Algorithm 1. Full CS is a naive version which does not combine contexts. (b) Time usage of each rule with error bars showing the standard error.

**Theorem 5** (Soundness). *Algorithm 1 always terminates: if it returns an expression, it is correct.*

In the setting of standard do-calculus, where no labels are present (in addition to those defining interventions) the search is complete for (conditional) causal effect identifiability. This is because the separation condition is general enough to capture all conditional independences used by do-calculus as shown by Theorem 2.

## 6 Experiments and Examples

We implemented the search in C++ and the code is available in the R-package dosearch on CRAN [31]. First we will present a simulation study on the search and then show a host of examples where identifiability can be shown with our approach. Experiments were performed on a modern desktop computer (single thread, Intel Core i7-4790, 3.4 GHz).

We considered DAGs with $n = 7, 8, 9$ nodes with 100 DAGs for each $n$. Edges for the DAGs were sampled randomly with average degree of 3. We sampled labels on the edges (local CSIs) with probability 0.5. Two of the nodes were considered latent and the aim was to determine whether $P(Y \mid \mathrm{do}(X))$ can be identified. Fig. 5(a) shows the running times of Algorithm 1 with a 30 minute timeout. The search times when all contexts are considered separate (i.e., the terms have fixed assigned values for all variables) are included as a baseline (full CS). Using terms that combine assignments as formulated in CSI-calculus considerably speeds up the execution times.

In the same simulation, we examined the effect of applying the individual rules on the total running times, as shown in Fig. 5(b). Rules 1 and 4 dominate the running time. For rule 1, considerable time is spent on checking whether the conditional independence constraints hold (recall that this step is also (co)NP-hard). Rule 4 combines two previously identified terms, and therefore a single term may help to identify further terms in a large number of ways.

Importantly, the search implementing CSI-calculus can prove identifiability of $P(Y \mid \mathrm{do}(X))$ for the LDAGs in Fig. 6 which would be non-identifiable otherwise via standard do-calculus. Non-identifiability can be verified by running ID on the underlying DAGs without labels or by noting that each graph includes a hedge. In Fig. 6(a) $P(Y \mid \mathrm{do}(X)) = P(Y \mid X, W = 1)$. Intuitively, node $W$ acts similarly as an intervention node and hence conditioning on $W = 1$ eliminates the back-door path. In Fig. 6(b) $P(Y \mid \mathrm{do}(X)) = P(Y)$, because $X$ and $Y$ are independent when $X$ is intervened on due to the labels. In Fig. 6(c) $P(Y \mid \mathrm{do}(X)) = P(Y \mid Z = 0, X)P(Z = 0) + P(Y \mid Z = 1)P(Z = 1)$, adjusting for $Z$ is needed, which opens up a new d-connecting path through $H$ and $Q$. Fortunately, when $Z = 0$ there is no confounding path, and when $Z = 1$ there is a confounding path but no directed path from $Z$. In Fig. 6(d), the causal effect is identifiable and the output by Algorithm 1 is:

$$P(Y \mid \mathrm{do}(X)) = P(A = 1){\textstyle\sum_W} P(Y \mid X, W, A = 1)P(W \mid A = 1)$$
$$+ P(A = 0){\textstyle\sum_Z} P(Z \mid X, A = 0){\textstyle\sum_{X'}} P(Y \mid X', Z, A = 0)P(X' \mid A = 0)$$

When $A = 1$, the first term resembles the back-door formula, adjusting for $W$. When $A = 0$, the second term resembles the front-door formula through $Z$. Since $A \perp\!\!\!\perp X, I_X$ in the LDAG, we are

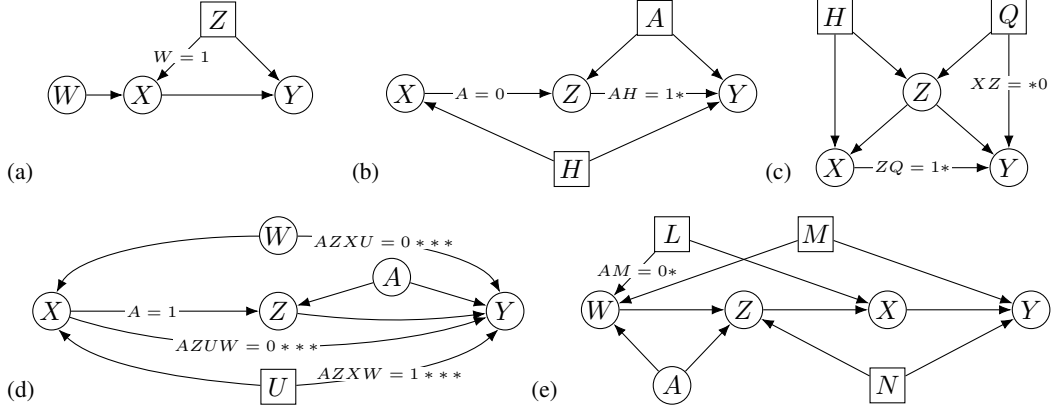

Figure 6: LDAGs such that $P(Y \mid do(X))$ is identifiable using CSIs, but not with standard do-calculus.

able to combine the formulas. In Fig. 6(e) when $A = 0$, the confounding path from $Y$ to $I_X$ vanishes allowing for a back-door type formula $P(Y \mid do(X)) = \sum_Z P(Z \mid A = 0) P(Y \mid X, Z, A = 0)$.

## 7   Discussion and Conclusion

In this paper, we considered causal effect identifiability in the presence of context-specific independence relations, which commonly arise from causal mechanisms over discrete variables. We formalized the problem employing LDAGs, showed that deciding causal effect non-identifiability is NP-hard when CSIs are present, developed a calculus, and designed a readily usable automatic search procedure for finding identifying formulas. We showed that with only a few additional CSIs, our approach may enable identifiability in previously non-identifiable cases.

Currently, we are at the level of a calculus and a search procedure over the calculus. Although the presented rules and the search are sound, completeness results are harder to obtain. Despite that the general decision problem is NP-hard, one could think of applying polynomial ID over context-specific DAGs and then combining the results in order to obtain a complete decision procedure. However, the following theorem shows that identifiability in context-specific DAGs is not a direct indicator of general identifiability.

**Theorem 6.** *Causal effect $P(Y \mid do(X))$ may be non-identifiable from $P(\boldsymbol{W})$ even if $P(Y \mid do(X))$ is identifiable in the context $\boldsymbol{s}$ specific DAGs for every $\boldsymbol{s} \in val(\boldsymbol{S})$ or if $P(Y \mid do(X), \boldsymbol{s})$ is identifiable in the context $\boldsymbol{s}$ specific DAGs for every $\boldsymbol{s} \in val(\boldsymbol{S})$ where $\boldsymbol{S}$ contains only observed variables.*

Hence further research is needed for a similar theory as for do-calculus, which resulted in completeness proofs through hedges, ID and IDC algorithms [28, 17, 27], if it is possible here. The generalization to categorical variables is mostly imminent, but designing a feasible search procedure is certainly an additional challenge. As such, the presented approach can already leverage from interventional distributions [3] by modifying the set of inputs $\boldsymbol{I}$ of Algorithm 1.

We believe our approach using CSIs will have an impact on a variety of related problems. We would like to use our approach to solve cases of transportability, selection bias and missing data problems [4, 5, 21]. The methodology presented is likely to render causal effects and distributions identifiable also in these problems, provided that there are CSI relations present.

### Acknowledgments

ST was supported by Academy of Finland grant 311877 (Decision analytics utilizing causal models and multiobjective optimization). AH was supported by Academy of Finland grant 295673.

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
