[Supplementary Material · supplement.pdf]

# Identifying Causal Effects
# via Context-specific Independence Relations
# *(Supplementary Material)*

**Santtu Tikka**
Department of Mathematics and Statistics
University of Jyvaskyla, Finland
`santtu.tikka@jyu.fi`

**Antti Hyttinen**
HIIT, Department of Computer Science
University of Helsinki, Finland
`antti.hyttinen@helsinki.fi`

**Juha Karvanen**
Department of Mathematics and Statistics
University of Jyvaskyla, Finland
`juha.t.karvanen@jyu.fi`

## A  Proofs

*Proof of Theorem 1 without positivity.* For pedagogical purposes, we begin with a proof without the positivity assumption. The subsequent proof, where positivity is assumed, is obtained via a slight modification.

Define the corresponding LDAG to the 3-SAT instance as follows (see Fig. 1) [2]. Let nodes $\boldsymbol{Z} = \{Z_1, \cdots, Z_l\}$ correspond to variables in the 3-SAT instance and $U_1, \cdots, U_k$ represent the clauses of the 3-SAT instance. Let the parents of node $U_i$ ($i \geq 1$) be $U_{i-1}$, and the $Z$s appearing in $i$th clause. The label $\boldsymbol{L}_i$ on the edge $U_{i-1} \to U_i$ consists of assignments to other parents of $U_i$ that are in $\boldsymbol{Z}$: let it be exactly the set of assignments that *do not* satisfy the $i$th clause. Let the parent of $X$ be $U_0$ and the parents of $Y$ be $X, U_k$. We will show that $P(Y \,|\, \mathrm{do}(X))$ is identifiable from $P(X, Y)$ if and only if the 3-SAT instance is unsatisfiable.

**Identifiability when UNSAT**   Suppose the 3-SAT instance is unsatisfiable. Then for any assignment $\boldsymbol{z}$ to $\boldsymbol{Z}$, edge $U_{i-1} \to U_i$ is absent in the context $\boldsymbol{z}$, and according to CSI-separation: $I_X \perp\!\!\!\perp Y \,|\, X, \boldsymbol{Z} = \boldsymbol{z}$. It follows that $I_X \perp\!\!\!\perp Y \,|\, X, \boldsymbol{Z}$. By d-separation $I_X \perp\!\!\!\perp \boldsymbol{Z}$, by contraction $I_X \perp\!\!\!\perp Y, \boldsymbol{Z} \,|\, X$, by decomposition $I_X \perp\!\!\!\perp Y \,|\, X$. Thus the causal effect is identifiable:

$$P(Y \,|\, \mathrm{do}(X)) = P(Y \,|\, X, I_X = 1) = P(Y \,|\, X, I_X = 0) = P(Y \,|\, X).$$

**Non-identifiability when SAT**   Suppose the 3-SAT instance is satisfiable. We define 2 parameterizations for the LDAG, $M^1$ and $M^2$ that agree with $P(X, Y)$ but disagree on $P(Y \,|\, \mathrm{do}(X))$. First let us describe the common part of both models. Let $\boldsymbol{Z}$ be distributed uniformly mutually independently. Let $U_i = U_{i-1}$ if $Z_a, Z_b, Z_c$ satisfy the clause $i$ and 0 otherwise. Let $U_0$ be uniform, and let $X$ be equal to $U_0$.

**Same observed distribution** We can calculate the observed joint $P(X, Y)$ as follows:

$$
\begin{aligned}
P(X, Y) &= \sum_{U_k, U_0, \boldsymbol{Z}} P(X, Y, U_k, U_0, \boldsymbol{Z}) \\
&= \sum_{U_k, U_0, \boldsymbol{Z}} P(Y \mid X, U_k) P(X \mid U_0) P(U_k \mid U_0, \boldsymbol{Z}) P(U_0) P(\boldsymbol{Z}) \quad \| P(\boldsymbol{Z}) = 2^{-l}, P(U_0) = 2^{-1} \\
&= 2^{-l-1} \sum_{U_k, U_0, \boldsymbol{Z}} P(Y \mid X, U_k) P(X \mid U_0) P(U_k \mid U_0, \boldsymbol{Z}) \quad \| P(X \mid U_0 = 1 - X) = 0 \\
&= 2^{-l-1} \sum_{U_k, \boldsymbol{Z}} P(Y \mid X, U_k) P(U_k \mid U_0 = X, \boldsymbol{Z}) \\
&= 2^{-l-1} \sum_{U_k, \boldsymbol{z} \in \text{SAT}} P(Y \mid X, U_k) P(U_k \mid U_0 = X, \boldsymbol{z}) \quad \| U_k = U_0 = X \text{ when SAT} \\
&\quad + 2^{-l-1} \sum_{U_k, \boldsymbol{z} \in \text{UNSAT}} P(Y \mid X, U_k) P(U_k \mid U_0 = X, \boldsymbol{z}) \quad \| U_k = 0 \text{ when UNSAT} \\
&= 2^{-l-1} \left[ \sum_{\boldsymbol{z} \in \text{SAT}} P(Y \mid X, U_k = X) + \sum_{\boldsymbol{z} \in \text{UNSAT}} P(Y \mid X, U_k = 0) \right].
\end{aligned}
$$

In order for this to be equal for $M^1$ and $M^2$ it suffices that

$$
\begin{aligned}
P^1(Y \mid X, U_k = 0) &= P^2(Y \mid X, U_k = 0), \\
P^1(Y \mid X = 1, U_k = 1) &= P^2(Y \mid X = 1, U_k = 1).
\end{aligned}
$$

In addition, the probabilities should be non-zero for $P(X, Y)$ to be positive for all 3-SAT instances. Then

$$
P^1(X, Y) = P^2(X, Y) > 0.
$$

**Different causal effects** The parameter $P(Y \mid X = 0, U_k = 1)$ does not affect the passive observational distribution in any way, so let $P^1(Y \mid X = 0, U_k = 1) \neq P^2(Y \mid X = 0, U_k = 1)$. This difference alters the causal effect:

$$
\begin{aligned}
P(Y \mid \text{do}(X = 0)) &= 2^{-l-1} \sum_{U_k, U_0, \boldsymbol{Z}} P(Y \mid X = 0, U_k) P(U_k \mid U_0, \boldsymbol{Z}) \\
&= 2^{-l-1} \sum_{U_k, U_0, \boldsymbol{z} \in \text{SAT}} P(Y \mid X = 0, U_k) P(U_k \mid U_0, \boldsymbol{z}) \quad \| U_k = U_0 \text{ when SAT} \\
&\quad + 2^{-l-1} \sum_{U_k, U_0, \boldsymbol{z} \in \text{UNSAT}} P(Y \mid X = 0, U_k) P(U_k \mid U_0, \boldsymbol{z}) \quad \| U_k = 0 \text{ when UNSAT} \\
&= 2^{-l-1} \sum_{\boldsymbol{z} \in \text{SAT}} P(Y \mid X = 0, U_k = U_0) + 2^{-l-1} \sum_{\boldsymbol{z} \in \text{UNSAT}} P(Y \mid X = 0, U_k = 0).
\end{aligned}
$$

and thus

$$
\begin{aligned}
P^1(Y \mid \text{do}(X = 0)) - P^2(Y \mid \text{do}(X = 0)) \quad &= \\
2^{-l-1} \sum_{\boldsymbol{Z} \in \text{SAT}} [P^1(Y \mid X = 0, U_k = 1) - P^2(Y \mid X = 0, U_k = 1)] \quad &\neq \quad 0.
\end{aligned}
$$

This shows that causal effect is non-identifiable if the 3-SAT instance is satisfiable. Next, we provide a similar construction for the case where positivity is assumed. $\square$

*Proof of Theorem 1.* Define the corresponding LDAG to the 3-SAT instance as follows (see Fig. 1) [2]. Let nodes $\boldsymbol{Z} = \{Z_1, \cdots, Z_l\}$ correspond to variables in the 3-SAT instance and $U_1, \cdots, U_k$ represent the clauses of the 3-SAT instance. Let the parents of node $U_i$ ($i \geq 1$) be $U_{i-1}$, and the $Z$s appearing in $i$th clause. The label $\boldsymbol{L}_i$ on the edge $U_{i-1} \to U_i$ consists of assignments to other parents of $U_i$ that are in $\boldsymbol{Z}$: let it be exactly the set of assignments that *do not* satisfy the $i$th clause. Let the parent of $X$ be $U_0$ and the parents of $Y$ be $X, U_k$. We will show that $P(Y \mid \text{do}(X))$ is identifiable from $P(X, Y)$ if and only if the 3-SAT instance is unsatisfiable.

**Identifiability when UNSAT** Suppose the 3-SAT instance is unsatisfiable. Then for any assignment $\boldsymbol{z}$ to $\boldsymbol{Z}$, edge $U_{i-1} \to U_i$ is absent in the context $\boldsymbol{z}$, and according to CSI-separation: $I_X \perp\!\!\!\perp Y \mid X, \boldsymbol{Z} = \boldsymbol{z}$. It follows that $I_X \perp\!\!\!\perp Y \mid X, \boldsymbol{Z}$. By d-separation $I_X \perp\!\!\!\perp \boldsymbol{Z}$, by contraction $I_X \perp\!\!\!\perp Y, \boldsymbol{Z} \mid X$, by decomposition $I_X \perp\!\!\!\perp Y \mid X$. Thus the causal effect is identifiable:

$$P(Y \mid \mathrm{do}(X)) = P(Y \mid X, I_X = 1) = P(Y \mid X, I_X = 0) = P(Y \mid X).$$

**Non-identifiability when SAT** Suppose the 3-SAT instance is satisfiable. We define 2 parameterizations for the LDAG, $M^1$ and $M^2$ that agree on $P(X, Y)$ but disagree on $P(Y \mid \mathrm{do}(X))$.

First let us describe the common part of both models. Let $\boldsymbol{Z}$ be distributed uniformly mutually independently. If $Z_a, Z_b, Z_c$ do not satisfy $i$:th clause we let $U_i$ to be distributed uniformly. If they do let $P(U_i = 1 \mid U_{i-1} = 1) = p$ and $P(U_i = 1 \mid U_{i-1} = 0) = 1 - p$ such that $p \neq 1/2$. For $M^1$ let $P^1(U_0 = 1) = 1/2$ and let $P^1(X = 1 \mid U_0 = 1) = q$ and $P^1(X = 1 \mid U_0 = 0) = 1 - q$ such that $q \neq 1/2$. For $M^2$ let $P^2(U_0 = 0) = q$ and $P^2(X \mid U_0) = 1/2$ irrespective of the value of $U_0$.

First we observe that $P^1(X) = P^2(X)$. For $M^1$ we have

$$P^1(X = 0) = P^1(X = 0 \mid U_0 = 0)P^1(U_0 = 0) + P^1(X = 0 \mid U_0 = 1)P^1(U_0 = 1)$$
$$= q\frac{1}{2} + (1 - q)\frac{1}{2} = \frac{1}{2},$$

and for $M^2$ we have

$$P^2(X = 0) = P^2(X = 0 \mid U_0 = 0)P^2(U_0 = 0) + P^2(X = 0 \mid U_0 = 1)P^2(U_0 = 1)$$
$$= \frac{1}{2}q + \frac{1}{2}(1 - q) = \frac{1}{2}.$$

If some clause $j$ is unsatisfied, the distributions all up to $k$ are uniform regardless whether the remaining clauses are satisfied or not:

$$P(U_j \mid U_0 = 1, \boldsymbol{z}) = \sum_{U_{j-1}} P(U_j \mid U_{j-1}, \boldsymbol{z})P(U_{j-1} \mid U_0 = 1, \boldsymbol{z}) = \frac{1}{2}\sum_{U_{j-1}} P(U_{j-1}|U_0, \boldsymbol{z}) = \frac{1}{2}.$$
$$P(U_{j+1} \mid U_0 = 1, \boldsymbol{z}) = \sum_{U_j} P(U_{j+1} \mid U_j, \boldsymbol{z})P(U_j \mid U_0 = 1, \boldsymbol{z}) = (1 - p)\frac{1}{2} + p\frac{1}{2} = \frac{1}{2}.$$

If all clauses are satisfied by $\boldsymbol{z}$ we can define the following random variable:

$$F = \sum_{i=1}^{k} I(U_i \neq U_{i-1}, \boldsymbol{z} \in \mathrm{SAT}),$$

where $I$ is the indicator function. Based on the parametrization of the $U_i$:s we know that $P(F \mid \boldsymbol{z}) = \mathrm{Bin}(k, 1 - p)$. Furthermore, $U_k = U_0$ if $F = 2t$ for any $t \in \{0, \dots, \lfloor k/2 \rfloor\}$. Similarly, $U_k \neq U_0$ if $F = 2t + 1$ for any $t \in \{0, \dots, \lfloor \frac{k-1}{2} \rfloor\}$. From this we obtain the following distributions

$$P(U_k = 0 \mid U_0 = 0, \boldsymbol{z}) = P(U_k = 1 \mid U_0 = 1, \boldsymbol{z}) = \sum_{t=0}^{\lfloor k/2 \rfloor} \binom{k}{2t}(1 - p)^{2t}p^{k-2t} := a.$$

$$P(U_k = 1 \mid U_0 = 0, \boldsymbol{z}) = P(U_k = 0 \mid U_0 = 1, \boldsymbol{z}) = \sum_{t=0}^{\lfloor \frac{k-1}{2} \rfloor} \binom{k}{2t+1}(1 - p)^{2t+1}p^{k-2t-1} = 1 - a.$$

**Same observed distribution**   We can calculate the observed joint $P(X, Y)$ for both models as follows:

$$P(X, Y) = \sum_{U_k, U_0, \boldsymbol{Z}} P(X, Y, U_k, U_0, \boldsymbol{Z})$$

$$= \sum_{U_k, U_0, \boldsymbol{Z}} P(Y \mid X, U_k) P(X \mid U_0) P(U_k \mid U_0, \boldsymbol{Z}) P(U_0) P(\boldsymbol{Z}) \quad \| \ P(\boldsymbol{Z}) = 2^{-l}$$

$$= 2^{-l} \sum_{U_k, U_0, \boldsymbol{Z}} P(Y \mid X, U_k) P(X \mid U_0) P(U_k \mid U_0, \boldsymbol{Z}) P(U_0)$$

$$= 2^{-l} \sum_{U_k, U_0, \boldsymbol{z} \in \text{SAT}} P(Y \mid X, U_k) P(X \mid U_0) P(U_k \mid U_0, \boldsymbol{z}) P(U_0)$$

$$+ \ 2^{-l} \sum_{U_k, U_0, \boldsymbol{z} \in \text{UNSAT}} P(Y \mid X, U_k) P(X \mid U_0) P(U_k \mid U_0, \boldsymbol{z}) P(U_0).$$

For $M^1$ and $M^2$ we have

$$P(X = 0, Y)$$

$$= 2^{-l} \sum_{\boldsymbol{z} \in \text{SAT}} \left( P(Y \mid X = 0, U_k = 0) q a \frac{1}{2} + P(Y \mid X = 0, U_k = 0)(1 - q)(1 - a)\frac{1}{2} \right.$$

$$\left. + \ P(Y \mid X = 0, U_k = 1) q (1 - a)\frac{1}{2} + P(Y \mid X = 0, U_k = 1)(1 - q)a\frac{1}{2} \right)$$

$$+ \ 2^{-l} \sum_{\boldsymbol{z} \in \text{UNSAT}} \left( P(Y \mid X = 0, U_k = 0) q \frac{1}{4} + P(Y \mid X = 0, U_k = 0)(1 - q)\frac{1}{4} \right.$$

$$\left. + \ P(Y \mid X = 0, U_k = 1) q \frac{1}{4} + P(Y \mid X = 0, U_k = 1)(1 - q)\frac{1}{4} \right)$$

$$= 2^{-l-1} \sum_{\boldsymbol{z} \in \text{SAT}} \left( P(Y \mid X = 0, U_k = 0) q a + P(Y \mid X = 0, U_k = 0)(1 - q)(1 - a) \right.$$

$$\left. + \ P(Y \mid X = 0, U_k = 1) q (1 - a) + P(Y \mid X = 0, U_k = 1)(1 - q)a \right)$$

$$+ \ 2^{-l-2} \sum_{\boldsymbol{z} \in \text{UNSAT}} \left( P(Y \mid X = 0, U_k = 0) + P(Y \mid X = 0, U_k = 1) \right).$$

When $X = 1$, the expression for $P(X = 1, Y)$ is different for the models. For $M^1$ it is

$$P^1(X = 1, Y)$$

$$= 2^{-l} \sum_{\boldsymbol{z} \in \text{SAT}} \left( P^1(Y \mid X = 1, U_k = 0)(1 - q)a\frac{1}{2} + P^1(Y \mid X = 1, U_k = 0) q (1 - a)\frac{1}{2} \right.$$

$$\left. + \ P^1(Y \mid X = 1, U_k = 1)(1 - q)(1 - a)\frac{1}{2} + P^1(Y \mid X = 1, U_k = 1) q a\frac{1}{2} \right)$$

$$+ \ 2^{-l} \sum_{\boldsymbol{z} \in \text{UNSAT}} \left( P^1(Y \mid X = 1, U_k = 0)(1 - q)\frac{1}{4} + P^1(Y \mid X = 1, U_k = 0) q \frac{1}{4} \right.$$

$$\left. + \ P^1(Y \mid X = 1, U_k = 1)(1 - q)\frac{1}{4} + P^1(Y \mid X = 1, U_k = 1) q \frac{1}{4} \right)$$

$$= 2^{-l-1} \sum_{\boldsymbol{z} \in \text{SAT}} \left( P^1(Y \mid X = 1, U_k = 0)(1 - q)a + P^1(Y \mid X = 1, U_k = 0) q (1 - a) \right.$$

$$\left. + \ P^1(Y \mid X = 1, U_k = 1)(1 - q)(1 - a) + P^1(Y \mid X = 1, U_k = 1) q a \right)$$

$$+ \ 2^{-l-2} \sum_{\boldsymbol{z} \in \text{UNSAT}} \left( P^1(Y \mid X = 1, U_k = 0) + P^1(Y \mid X = 1, U_k = 1) \right),$$

and for $M^2$ it is

$$P^2(X = 1, Y)$$

$$= 2^{-l} \sum_{\mathbf{z} \in \text{SAT}} \left( P^2(Y \mid X = 1, U_k = 0)qa\frac{1}{2} + P^2(Y \mid X = 1, U_k = 0)(1 - q)(1 - a)\frac{1}{2} \right.$$

$$\left. + P^2(Y \mid X = 1, U_k = 1)q(1 - a)\frac{1}{2} + P^2(Y \mid X = 1, U_k = 1)(1 - q)a\frac{1}{2} \right)$$

$$+ 2^{-l} \sum_{\mathbf{z} \in \text{UNSAT}} \left( P^2(Y \mid X = 1, U_k = 0)q\frac{1}{4} + P^2(Y \mid X = 1, U_k = 0)(1 - q)\frac{1}{4} \right.$$

$$\left. + P^2(Y \mid X = 1, U_k = 1)q\frac{1}{4} + P^2(Y \mid X = 1, U_k = 1)(1 - q)\frac{1}{4} \right)$$

$$= 2^{-l-1} \sum_{\mathbf{z} \in \text{SAT}} \left( P^2(Y \mid X = 1, U_k = 0)qa + P^2(Y \mid X = 1, U_k = 0)(1 - q)(1 - a) \right.$$

$$\left. + P^2(Y \mid X = 1, U_k = 1)q(1 - a) + P^2(Y \mid X = 1, U_k = 1)(1 - q)a \right)$$

$$+ 2^{-l-2} \sum_{\mathbf{z} \in \text{UNSAT}} (P^2(Y \mid X = 1, U_k = 0) + P^2(Y \mid X = 1, U_k = 1)).$$

In order to ensure that $P^1(X = 1, Y) = P^2(X = 1, Y)$ we let $P^1(Y \mid X = 1, U_k) = P^2(Y \mid X = 1, U_k) = 1/2$. For $P(Y \mid X = 0, U_k)$ we let $P^1(Y \mid X = 0, U_k) = P^2(Y \mid X = 0, U_k) > 0$. The considered parametrization ensures that $P(X, Y, \mathbf{Z}, U_0, \ldots, U_k)$ is positive for all 3-SAT instances, and

$$P^1(X, Y, \mathbf{Z}, U_0, \ldots, U_k) = P^2(X, Y, \mathbf{Z}, U_0, \ldots, U_k) > 0.$$

**Different causal effects**  The different parameterizations for $U_0$ and the conditional distribution $P(X \mid U_0)$ in $M^1$ and $M^2$ induce a difference in the causal effects. The causal effect can be computed as

$$P(Y \mid \text{do}(X = 0))$$

$$= 2^{-l} \sum_{U_k, U_0, \mathbf{z} \in \text{SAT}} P(Y \mid X, U_k)P(U_k \mid U_0, \mathbf{z})P(U_0)$$

$$+ 2^{-l} \sum_{U_k, U_0, \mathbf{z} \in \text{UNSAT}} P(Y \mid X, U_k)P(U_k \mid U_0, \mathbf{z})P(U_0).$$

For model $M^1$ this is

$$P^1(Y \mid \text{do}(X = 0))$$

$$= 2^{-l} \sum_{\mathbf{z} \in \text{SAT}} \left( P^1(Y \mid X = 0, U_k = 0)a\frac{1}{2} + P^1(Y \mid X = 0, U_k = 0)(1 - a)\frac{1}{2} \right.$$

$$\left. + P^1(Y \mid X = 0, U_k = 1)(1 - a)\frac{1}{2} + P^1(Y \mid X = 0, U_k = 1)a\frac{1}{2} \right)$$

$$+ 2^{-l} \sum_{\mathbf{z} \in \text{UNSAT}} \left( P^1(Y \mid X = 0, U_k = 0)\frac{1}{4} + P^1(Y \mid X = 0, U_k = 0)\frac{1}{4} \right.$$

$$\left. + P^1(Y \mid X = 0, U_k = 1)\frac{1}{4} + P^1(Y \mid X = 0, U_k = 1)\frac{1}{4} \right)$$

$$= 2^{-l-1} \sum_{\mathbf{z} \in \text{SAT}} (P^1(Y \mid X = 0, U_k = 0) + P^1(Y \mid X = 0, U_k = 1)$$

$$+ 2^{-l-1} \sum_{\mathbf{z} \in \text{UNSAT}} (P^1(Y \mid X = 0, U_k = 0) + P^1(Y \mid X = 0, U_k = 1)).$$

Figure 1: LDAG for the proof of Theorem 1.

For model $M^2$ the causal effect is

$$P^2(Y \mid \mathrm{do}(X = 0))$$

$$= 2^{-l} \sum_{z \in \mathrm{SAT}} \big( P^2(Y \mid X = 0, U_k = 0)aq + P^2(Y \mid X = 0, U_k = 0)(1-a)(1-q)$$

$$+ P^2(Y \mid X = 0, U_k = 1)(1-a)q + P^2(Y \mid X = 0, U_k = 1)a(1-q) \big)$$

$$+ 2^{-l} \sum_{z \in \mathrm{UNSAT}} \bigg( P^2(Y \mid X = 0, U_k = 0)\frac{1}{2}q + P^2(Y \mid X = 0, U_k = 0)\frac{1}{2}(1-q)$$

$$+ P^2(Y \mid X = 0, U_k = 1)\frac{1}{2}q + P^2(Y \mid X = 0, U_k = 1)\frac{1}{2}(1-q) \bigg)$$

$$= 2^{-l} \sum_{z \in \mathrm{SAT}} \big( P^2(Y \mid X = 0, U_k = 0)aq + P^2(Y \mid X = 0, U_k = 0)(1-a)(1-q)$$

$$+ P^2(Y \mid X = 0, U_k = 1)(1-a)q + P^2(Y \mid X = 0, U_k = 1)a(1-q) \big)$$

$$+ 2^{-l-1} \sum_{z \in \mathrm{UNSAT}} \left( P(Y \mid X = 0, U_k = 0) + P(Y \mid X = 0, U_k = 1) \right).$$

Thus

$$P^1(Y \mid \mathrm{do}(X = 0)) - P^2(Y \mid \mathrm{do}(X = 0))$$

$$= 2^{-l} \sum_{z \in \mathrm{SAT}} \left( \left( \frac{1}{2} - aq - (1-a)(1-q) \right) P(Y \mid X = 0, U_k = 0) \right.$$

$$\left. + \left( \frac{1}{2} - (1-a)q - a(1-q) \right) P(Y \mid X = 0, U_k = 1) \right)$$

$$= 2^{-l} \sum_{z \in \mathrm{SAT}} \left( r P(Y \mid X = 0, U_k = 0) - r P(Y \mid X = 0, U_k = 1) \right).$$

where $r = 1/2 - aq - (1-a)(1-q)$. It is apparent that we can always choose $q$ and $p$ (i.e., $a$) in such a way that the causal effects will differ as long as $P(Y \mid X = 0, U_k = 0) \neq P(Y \mid X = 0, U_k = 1)$. This shows that the causal effect is non-identifiable if the 3-SAT instance is satisfiable. $\qquad \square$

*Proof of Theorem 2.* Note that this proof follows closely to the original soundness proofs of do-calculus [3]. Any formula that is derivable with do-calculus is also derivable here with rule 1 of the calculus presented in the main paper. This is because conditioning on $I_X = 1$ is exactly equivalent to removing all incoming edges of a node. The equivalent inference rules to the do-calculus formulation in [1] are presented below, the notation $\| X$ means that $X$ are intervened and all edges into them are cut.

Rule 1 (insertion/deletion of observations) originally

$$P(Y \mid \mathrm{do}(X), Z, W) = P(Y \mid \mathrm{do}(X), W) \text{ if } Y \perp\!\!\!\perp Z \mid X, W \| X$$

is just

$$P(Y \mid X, I_X = 1, Z, W) = P(Y \mid X, I_X = 1, W) \text{ if } Y \perp\!\!\!\perp Z \mid X, W, I_X = 1$$

Rule 2 (action/observation exchange) originally

$$P(\boldsymbol{Y}\,|\,\mathrm{do}(\boldsymbol{X}),\mathrm{do}(\boldsymbol{Z}),\boldsymbol{W}) = P(\boldsymbol{Y}\,|\,\mathrm{do}(\boldsymbol{X}),\boldsymbol{Z},\boldsymbol{W}) \text{ if } \boldsymbol{Y}\perp\!\!\!\perp \boldsymbol{I_Z}\,|\,\boldsymbol{X},\boldsymbol{Z},\boldsymbol{W}\,\|\,\boldsymbol{X}$$

is simply

$$P(\boldsymbol{Y}\,|\,\boldsymbol{X},\boldsymbol{I_X}=1,\boldsymbol{Z},\boldsymbol{I_Z}=1,\boldsymbol{W}) = P(\boldsymbol{Y}\,|\,\boldsymbol{X},\boldsymbol{I_X}=1,\boldsymbol{Z},\boldsymbol{I_Z}=0,\boldsymbol{W})$$
$$\text{if } \boldsymbol{Y}\perp\!\!\!\perp \boldsymbol{I_Z}\,|\,\boldsymbol{X},\boldsymbol{Z},\boldsymbol{W},\boldsymbol{I_X}=1$$

Rule 3 (Insertion/deletion of actions) originally in form

$$P(\boldsymbol{Y}\,|\,\mathrm{do}(\boldsymbol{X}),\mathrm{do}(\boldsymbol{Z}),\boldsymbol{W}) = P(\boldsymbol{Y}\,|\,\mathrm{do}(\boldsymbol{X}),\boldsymbol{W}) \text{ if } \boldsymbol{Y}\perp\!\!\!\perp \boldsymbol{I_Z}\,|\,\boldsymbol{X},\boldsymbol{W}\,\|\,\boldsymbol{X}$$

is a bit more complicated.

One way of deriving the equivalence with the presented rule is to first drop conditioning on $\boldsymbol{Z}$ and then change conditioning on $\boldsymbol{I_Z}=1$.

$$P(\boldsymbol{Y}\,|\,\boldsymbol{X},\boldsymbol{I_X}=1,\boldsymbol{Z},\boldsymbol{I_Z}=1,\boldsymbol{W}) = P(\boldsymbol{Y}\,|\,\boldsymbol{X},\boldsymbol{I_X}=1,\boldsymbol{I_Z}=1,\boldsymbol{W})$$
$$\text{if } \boldsymbol{Y}\perp\!\!\!\perp \boldsymbol{Z}\,|\,\boldsymbol{X},\boldsymbol{W},\boldsymbol{I_X}=1,\boldsymbol{I_Z}=1$$
$$P(\boldsymbol{Y}\,|\,\boldsymbol{X},\boldsymbol{I_X}=1,\boldsymbol{I_Z}=1,\boldsymbol{W}) = P(\boldsymbol{Y}\,|\,\boldsymbol{X},\boldsymbol{I_X}=1,\boldsymbol{I_Z}=0,\boldsymbol{W})$$
$$\text{if } \boldsymbol{Y}\perp\!\!\!\perp \boldsymbol{I_Z}\,|\,\boldsymbol{X},\boldsymbol{W},\boldsymbol{I_X}=1$$

In the lemma below we show the equivalence of the original condition to the conjunction of the new conditions. Anything that can be derived with rule 3 of do-calculus can also be derived with rule 1 of the paper. $\qquad\square$

**Lemma 1.** *D-separation relation* $\boldsymbol{Y}\perp\!\!\!\perp \boldsymbol{I_Z}\,|\,\boldsymbol{X},\boldsymbol{W}\,\|\,\boldsymbol{X}$ *is equivalent to* $[\boldsymbol{Y}\perp\!\!\!\perp \boldsymbol{I_Z}\,|\,\boldsymbol{X},\boldsymbol{I_X}=1,\boldsymbol{W}]\wedge[\boldsymbol{Y}\perp\!\!\!\perp \boldsymbol{Z}\,|\,\boldsymbol{X},\boldsymbol{W},\boldsymbol{I_X}=1,\boldsymbol{I_Z}=1].$

*Proof.* The original graphical condition $\boldsymbol{Y}\perp\!\!\!\perp \boldsymbol{I_Z}\,|\,\boldsymbol{X},\boldsymbol{W}\,\|\,\boldsymbol{X}$ is clearly equivalent to the first condition $\boldsymbol{Y}\perp\!\!\!\perp \boldsymbol{I_Z}\,|\,\boldsymbol{X},\boldsymbol{I_X}=1,\boldsymbol{W}$, the only difference is notational. This shows one direction and the first part of the other direction. What remains to be shown is that $\boldsymbol{Y}\perp\!\!\!\perp \boldsymbol{Z}\,|\,\boldsymbol{X},\boldsymbol{W},\boldsymbol{I_X}=1,\boldsymbol{I_Z}=1$ follows from $\boldsymbol{Y}\perp\!\!\!\perp \boldsymbol{I_Z}\,|\,\boldsymbol{X},\boldsymbol{W}\,\|\,\boldsymbol{X}$.

We are not assuming any other labels or CSIs on the graph except for the ones brought by the use of intervention variables. We assume the form of d-separation which allows for repeated edges [4]. We also leave out the intervening and conditioning on $X$ in both conditions as we can just delete edges into $X$ in all considered graphs. Proving here the contrapositive, so assume $\boldsymbol{Y}\not\perp\!\!\!\perp \boldsymbol{Z}\,|\,\boldsymbol{W},\boldsymbol{I_Z}=1$. Thus the is a d-connecting path between some $Y_j\in\boldsymbol{Y}$ and some $Z_i\in\boldsymbol{Z}$ active when conditioning on $\boldsymbol{W},\boldsymbol{I_Z}=1$. Due to the nature of intervention and intervention variables, conditioning on $\boldsymbol{I_Z}=1$ can only break d-connecting paths, so we can drop the conditioning on $\boldsymbol{I_Z}$. Then, continuing the path with the arc $I_{Z_i}\to Z_i$ gives us a d-connecting path between $I_{Z_i}$ and $Y_j$. It is not intercepted at $Z_i$, since $Z_i$ was originally intervened on ($I_{Z_i}=1$), and thus the path to $Y_j$ must be out of $Z_i$. Hence $\boldsymbol{Y}\not\perp\!\!\!\perp \boldsymbol{I_Z}\,|\,\boldsymbol{W}$. This shows that $\boldsymbol{Y}\perp\!\!\!\perp \boldsymbol{I_Z}\,|\,\boldsymbol{W}\ \Rightarrow\ \boldsymbol{Y}\perp\!\!\!\perp \boldsymbol{Z}\,|\,\boldsymbol{W},\boldsymbol{I_Z}=1$ and adding intervention on $X$ back in we have $\boldsymbol{Y}\perp\!\!\!\perp \boldsymbol{I_Z}\,|\,\boldsymbol{X},\boldsymbol{W}\,\|\,\boldsymbol{X}\ \Rightarrow\ \boldsymbol{Y}\perp\!\!\!\perp \boldsymbol{Z}\,|\,\boldsymbol{X},\boldsymbol{W},\boldsymbol{I_X}=1,\boldsymbol{I_Z}=1.$ $\qquad\square$

*Proof of Theorem 3.* Suppose that there exists a set $\boldsymbol{C}$ such that $\boldsymbol{Y}\perp\!\!\!\perp \boldsymbol{Z}\,|\,\boldsymbol{X},\boldsymbol{w},\boldsymbol{C}$ is implied by an LDAG $G$. If $\boldsymbol{Y}\perp\!\!\!\perp \boldsymbol{C}\,|\,\boldsymbol{X},\boldsymbol{w}$, then $\boldsymbol{Y}\perp\!\!\!\perp \boldsymbol{C},\boldsymbol{Z}\,|\,\boldsymbol{X},\boldsymbol{w}$ by contraction and $\boldsymbol{Y}\perp\!\!\!\perp \boldsymbol{Z}\,|\,\boldsymbol{X},\boldsymbol{w}$ by decomposition. If $\boldsymbol{C}\perp\!\!\!\perp \boldsymbol{Z}\,|\,\boldsymbol{X},\boldsymbol{w}$, the CSI is obtained similarly by deriving $\boldsymbol{Z}\perp\!\!\!\perp \boldsymbol{Y},\boldsymbol{C}\,|\,\boldsymbol{X},\boldsymbol{w}$ by contraction. If $\boldsymbol{Y}\perp\!\!\!\perp \boldsymbol{C}\,|\,\boldsymbol{X},\boldsymbol{Z},\boldsymbol{w}$ in $G$ then we can write

$$P(\boldsymbol{Y}\,|\,\boldsymbol{X},\boldsymbol{w}) = \sum_{\boldsymbol{C}} P(\boldsymbol{Y}\,|\,\boldsymbol{X},\boldsymbol{w},\boldsymbol{c})P(\boldsymbol{c}\,|\,\boldsymbol{X},\boldsymbol{w})$$
$$= \sum_{\boldsymbol{C}} P(\boldsymbol{Y}\,|\,\boldsymbol{X},\boldsymbol{Z},\boldsymbol{w},\boldsymbol{c})P(\boldsymbol{c}\,|\,\boldsymbol{X},\boldsymbol{w})$$
$$= \sum_{\boldsymbol{C}} P(\boldsymbol{Y}\,|\,\boldsymbol{X},\boldsymbol{Z},\boldsymbol{w})P(\boldsymbol{c}\,|\,\boldsymbol{X},\boldsymbol{w})$$
$$= P(\boldsymbol{Y}\,|\,\boldsymbol{X},\boldsymbol{Z},\boldsymbol{w}).$$

If $\boldsymbol{Z}\perp\!\!\!\perp \boldsymbol{C}\,|\,\boldsymbol{X},\boldsymbol{Y},\boldsymbol{w}$, the result follows by swapping the roles of $\boldsymbol{Y}$ and $\boldsymbol{Z}$ in the above derivation. $\qquad\square$

Figure 2: LDAGs for the proof of Theorem 6.

*Proof of Theorem 4.* Let $\boldsymbol{c}_i \in val(\boldsymbol{C})$ and let $\boldsymbol{R}_i$ be a representative of $[\boldsymbol{c}_i]_{\underset{\sim}{s}}$. Then $\boldsymbol{Y}$ and $\boldsymbol{Z}$ are CSI-separated by $\boldsymbol{X}$ in the context $\boldsymbol{w}, \boldsymbol{c}_j$ in $G$ for every $\boldsymbol{c}_j \in val(\boldsymbol{C})$ such that $[\boldsymbol{c}_j]_{\underset{\sim}{s}} = [\boldsymbol{c}_i]_{\underset{\sim}{s}}$ because their context-specific DAGs are identical. Since the choice of $\boldsymbol{c}_i$ is arbitrary and the equivalence classes partition $val(\boldsymbol{C})$, the claim holds for all $\boldsymbol{c} \in val(\boldsymbol{C})$. $\qquad\square$

*Proof of Theorem 5.* Each new term is identified by using only valid rules of the calculus described in the paper and by ensuring that the required CSIs are implied by $G$ when rule 1 is applied. Theorem 2 shows that CSI-separation is complete for determining CSIs resulting from intervention nodes. It follows that if the search terminates and returns a formula for the target distribution, it was reached from $P(\boldsymbol{W})$ through a chain of valid manipulations. Suppose now that the search returns NA. By definition, Algorithm 1 enumerates every rule in every valid way. Furthermore, the order in which the distributions are expanded is irrelevant when identifiability is concerned, since every possible manipulation will still be carried out. The search will eventually terminate, since we only identify any specific term once and the set of valid manipulations through the rules for any given term is clearly finite. $\qquad\square$

*Proof of Theorem 6.* First we show that identifiability of $P(Y \mid do(X))$ in every context specific DAG does not imply identifiability in the LDAG itself. Consider LDAG of Figure 2(a). Clearly for any context over $S = Q, W$ there are no latent confounders between $X$ and $Y$ and therefore the effect $P(Y \mid do(X))$ is identifiable for all context-specific DAGs.

Let unobserved $Q, W$ be uniformly distributed. Let $X := Q \wedge W$ with a flip with probability .1. Let $Y := Q \vee W$, with a flip with probability .1. The positive joint can be described by:

| $Q$ | $W$ | $X = 0$ | $X = 1$ | $Y = 0$ | $Y = 1$ |
|---|---|---|---|---|---|
| 0 | 0 | 9/10 | 1/10 | 9/10 | 1/10 |
| 0 | 1 | 9/10 | 1/10 | 1/10 | 9/10 |
| 1 | 0 | 9/10 | 1/10 | 1/10 | 9/10 |
| 1 | 1 | 1/10 | 9/10 | 1/10 | 9/10 |

Which induces marginal:

| $P(X, Y)$ | $Y = 0$ | $Y = 1$ |
|---|---|---|
| $X = 0$ | 1/4 | 9/20 |
| $X = 1$ | 1/20 | 1/4 |

The marginal distributions are $P(X) = (7/10, 3/10)$ and $P(Y) = (3/10, 7/10)$. Conditionals of $Y$ are $P(Y \mid X = 0) = (5/14, 9/14)$ and $P(Y \mid X = 1) = (1/6, 5/6)$. The causal effect $P(Y \mid do(X)) = P(Y)$ as forcing $X$ does not influence $Y$. Let the second model be such that $Q, W$ are uniform, $X$ is distributed as $P(X)$ in the first model, and $Y$ as $P(Y \mid X)$. The marginal of $X, Y$ is the same, labels are respected, but causal effect $P(Y \mid do(X)) = P(Y \mid X) \neq P(Y)$.

Next we show that identifiability of $P(Y \mid do(X), \boldsymbol{c})$ does not imply the identifiability of $P(Y \mid do(X))$ in the LDAG. Consider the LDAG of Fig. 2(b). It is possible to identify the conditional causal effects $P(Y \mid do(X), A = 0)$ and $P(Y \mid do(X), A = 1)$ from $P(X, Y, A)$ since

$$P(Y \mid do(X), A) = P(Y \mid A, X, I_X = 1) = P(Y \mid A, X),$$

as $Y$ is independent of $I_X$ in the LDAG. However, the causal effect $P(Y \mid do(X))$ is not identifiable. To show this, we define 2 parameterizations, $M^1$ and $M^2$, for the LDAG, that agree on $P(X, Y, A)$

but disagree on $P(Y \mid \text{do}(X))$. In the first model we let $P^1(U) = 1/2$, $P^1(X = 1 \mid U = 1) = q$ and $P(X = 1 \mid U = 0) = 1 - q$. In the second model we let $P^2(U = 1) = q$ and $P^2(X \mid U) = 1/2$. We also let $P^1(A \mid X = 1, U) = P^2(A \mid X = 1, U) = P(A \mid X = 1, U)$.

We have that

$$P(X, A) = \sum_U P(X, A, U)$$

$$= P(U = 1)P(X \mid U = 1)P(A \mid X, U = 1) + P(U = 1)P(X \mid U = 1)P(A \mid X, U = 1).$$

For $X = 1$ and we have

$$P^1(X = 1, A) = P^2(X = 1, A) = \frac{1}{2}qP(A \mid X = 1, U = 1) + \frac{1}{2}(1 - q)P(A \mid X = 1, U = 0),$$

for $X = 0$ we have

$$P^1(X = 1, A) = \frac{1}{2}(1 - q)P^1(A \mid X = 0, U = 1) + \frac{1}{2}qP^1(A \mid X = 0, U = 0),$$

$$P^2(X = 1, A) = \frac{1}{2}qP^2(A \mid X = 0, U = 1) + \frac{1}{2}(1 - q)P^2(A \mid X = 0, U = 0),$$

thus we set $P(A \mid X = 0, U = 1) = P(A \mid X = 0, U = 0) = 1/2$ for both $M^1$ and $M^2$. Furthermore, we let $P^1(Y, Z \mid, X, A) = P^2(Y, Z \mid X, A) = P(Y, Z \mid X, A)$. This parametrization produces the same observed joint distribution $P^1(Y, X, A) = P^2(Y, X, A)$. Computing the causal effects result in

$$P^1(Y \mid \text{do}(X = 1)) = \sum_{U, A, Z} P^1(U)P^1(A \mid U, X = 1)P^1(Y, Z \mid X = 1, A)$$

$$= \sum_{A, Z} \left( \frac{1}{2}P(A \mid X = 1, U = 1)P(Y, Z \mid X = 1, A) \right.$$

$$\left. + \frac{1}{2}P(A \mid X = 1, U = 0)P(Y, Z \mid X = 1, A) \right),$$

and

$$P^2(Y \mid \text{do}(X = 1)) = \sum_{U, A, Z} P^2(U)P^2(A \mid U, X = 1)P^2(Y, Z \mid X = 1, A)$$

$$= \sum_{A, Z} (qP(A \mid X = 1, U = 1)P(Y, Z \mid X = 1, A)$$

$$+ (1 - q)P(A \mid X = 1, U = 0)P(Y, Z \mid X = 1, A)).$$

The difference $P^1(Y \mid \text{do}(X = 1)) - P^2(Y \mid \text{do}(X = 1))$ is

$$\sum_{A, Z} \left[ P(Y, Z \mid X = 1, A) \left( \left( \frac{1}{2} - q \right) P(A \mid X = 1, U = 1) - \left( \frac{1}{2} - q \right) P(A \mid X = 1, U = 0) \right) \right],$$

which is non-zero as long as $P(A \mid X = 1, U = 1) \neq P(A \mid X = 1, U = 0)$.

$\square$

# B   Heuristics

In order to guide the search to identify the most promising terms, we relate the source distributions to the target quantity through a proximity function and always expand the closest term first. We construct our proximity function $h$ by comparing the variables and value assignments that appear on the left-hand and right-hand sides of the target $P^t = P(\boldsymbol{Y}_1, \boldsymbol{y}_2 \mid \boldsymbol{X}_1, \boldsymbol{x}_2)$ and the source term $P^s = P(\boldsymbol{A}_1, \boldsymbol{a}_2 \mid \boldsymbol{B}_1, \boldsymbol{b}_2)$. Let $\boldsymbol{S}$ be the set of variables where value assignments between the terms agree and let $\boldsymbol{T}$ be the complement where the values disagree. We also denote $\boldsymbol{Y} = \boldsymbol{Y}_1 \cup \boldsymbol{Y}_2$, $\boldsymbol{X} = \boldsymbol{X}_1 \cup \boldsymbol{X}_2$, $\boldsymbol{A} = \boldsymbol{A}_1 \cup \boldsymbol{A}_2$ and $\boldsymbol{B} = \boldsymbol{B}_1 \cup \boldsymbol{B}_2$. Following the approach of [5], we define the proximity function as:

$h(P^t, P^s) = 10|\boldsymbol{Y} \cap \boldsymbol{A}| + 5|\boldsymbol{X} \cap \boldsymbol{B}| + 3|\boldsymbol{S}| - 2|\boldsymbol{Y} \setminus \boldsymbol{A}| - 2|\boldsymbol{A} \setminus \boldsymbol{Y}| - 2|\boldsymbol{X} \setminus \boldsymbol{B}| - 2|\boldsymbol{B} \setminus \boldsymbol{X}| - |\boldsymbol{T}|.$

Note that we also penalize the term $|\boldsymbol{A} \setminus \boldsymbol{Y}|$ even though extra variables on the left-hand side of a term no do not hinder identifiability. Having a penalty for this term seems to have the tendency to produce simpler formulas for the causal effects.