[Reviews · NeurIPS 2019]

Reviewer 1



This paper proposes a new identification strategy in causal effect estimation on a graph, and the identification strategy generalizes the framework of do-calculus. This paper shows a novel algorithm for searching an identifiable causal effect, which is an NP-hard problem. Authors also show the experimental results and provide their code for reproducing the results. One of the drawbacks of this paper is the complicated notation. For example, authors should define the do-calculus explicitly in their paper even though it is a well-known concept in this field, and we can understand the meaning after reading several pages. In addition to the above drawback, I could not understand the intuition of the rules of calculus. I think that authors should provide a more intuitive explanation about the each rule to justify them.

Reviewer 2



The author for the first time formulated the problem of causal effect identifiability in the presence of CSIs for binary variables, which is original. The manuscript is well structured and easy to follow; more details could be given for 5.2. The proofs for the rules are with good quality. Some more simulation results could be provided for a better understanding of the proposed algorithm. There are several concerns: 1. In 4.1, have all the rules been listed? 2. Will the greedy algorithm recover all the identifiable probability distributions? Although the problem is NP hard, can you provide a comparison of the greedy search result with an exhaustive search in the simulation? 3. How to identify the representatives of val(C)/~^s ? Is it through exhaustive search? Update: Thanks for the author for answering the questions. I think it is still not very clear to me what is the gap between the set of all identifiable causal effects and the rules identifiable using the rules listed in 4.1. But overall it is a very interesting work with good quality and I will keep my initial score.

Reviewer 3



This paper proposes an automated search procedure for identifying causal effects when context-specific independence relations are present in an observed distribution. Equipped with sufficient conditions for conditional independence statements (Boutiller et al. 1996) and LDAG representation (Pensar et al. 2015), a simple search algorithm is implemented. Overall, the paper is clearly written, and it was easy to follow theorems (clarity). However, it is hard to measure the novelty of the paper (originality), which I will discuss below. Hence, the proposed algorithm may be useful for some researchers, but its significance (impact) is unclear. It is nice to see the rules (basic probability axioms + (CS) independence) written clearly, which lead to the implementation of a search algorithm. However, there is nothing special about the rules. A set of sufficient criteria is implemented to check them in a sufficiently fast way. Is necessary (and sufficient) criteria impossible to provide even with a faithfulness assumption (or its modified version taking CSI into consideration)? The primary contribution, perceived and highlighted by the authors, seems to be the implementation of the search algorithm. However, the implementation cannot usually be an important factor to weigh in the acceptance of the paper. ** after rebuttal ** I somewhat disagree with the authors rebuttal in L39–41. Do-calculus *is* the result of basic probability axioms and conditional independence relations in a causal graph. (The authors proved it using LDAG w/ interventional nodes for Thm 2 in the appendix.) There is no reason to “prevent the need for … do-calculus.” But, I generally agree with the authors’ intention. Anyhow, I value the importance of the paper, and recommend the acceptance of the paper.

[Author Response · NeurIPS 2019]

**Thank you for your reviews and the excellent constructive feedback!**

**Reviewer 1**

The intuition of the rules is provided in their titles. We agree that further intuitive explanation of them is in order and
we will gladly add this. In particular:

- Rule 1 follows from the definition of CSI which includes CI as a special case.
- Marginalization, conditioning and factorization from standard probability calculus are operationalized by rules
2–4, respectively.
- Rule 5 uses the law of total probability to obtain the probability of the complement.
- Rule 6 explicates that if we know the expression for each assignment $Z = z$ then we also know the expression
without a specific assignment for $Z$.
- Rules 7 and 8 formulate the fact that if an expression is known for all assignments to $Z$, it is known for a
specific assignment $Z = z$.

We will add the rules of do-calculus into the paper to make it more self-sufficient. Note that due to Theorem 2 we do
not explicitly need to use them in our approach.

Could the reviewer explicitly point out what is meant by "some variables" that need more explicit definition and what
"parameters of interest" need explicit explanation, so that we could remedy this issue? We would also appreciate explicit
pointers to where the paper "has complicated notation", as reviewers 2 and 3 label it as "easy to follow".

**Reviewer 2**

Yes, Section 4.1 lists all the rules that are in use.

We would like to note that "greedy" is a somewhat misleading description of our algorithm (we do not use this term in
the paper). We do always first expand the term that our heuristic judges to be closest to the target. When the target is
identifiable, we stop when the target is first derived. The goal of the heuristic is to reach identifiability faster when the
target is identifiable. However, when the target is non-identifiable, we need to expand enough terms such that we can
guarantee that the target cannot be reached. If we were to expand *only* the closest term to the target greedily, several
identifiable instances would be left non-identified because the formulas and derivations are highly non-trivial (see e.g.
the several paths leading from the input to the target in Figure 3). We will explain this in the paper. We shall also try to
find ways to study greedy and exhaustive behavior in simulations, any more detailed suggestions are welcome.

In the implementation, representatives of $val(C)/\overset{s}{\sim}$ are obtained by going through the assignments of $C$ in lexico-
graphic order. The representative of a context-specific DAG is the value assignment that was evaluated first. There is no
need to specifically perform a search to obtain the representatives.

**Reviewer 3**

Implementation is not the primary contribution as the reviewer suggests in the detailed comments. We formulate the
important and relevant previously unconsidered problem, show computational NP-hardness result and solve the problem
by designing a calculus and a search algorithm. We identify causal effects (using CSIs) when previous algorithms can't
do so. We think that the whole methodology is a significant contribution and there is impact e.g. in the extensions to
transportability and selection bias. We will reformulate the beginning of Section 7 to highlight these (similarly as in the
abstract).

We spent a considerable amount of effort and time in finding and formulating the set of rules we use. Note that the
current set is special enough e.g. to prevent the need for the 3 arguably special rules of do-calculus. Each rule is
necessary, there are identifiable instances that are cannot be identified when excluding any single rule. We will further
highlight these points in the paper.

How would CSI-faithfulness (with regard to CIs and CSIs due to labels) help in getting a sufficient and necessary
separation criterion? For DAGs, d-separation is a sufficient and necessary condition for CIs implied by the structure.
CI-faithfulness then assumes that no additional CIs are present due to the parameters. A necessary condition for CSIs
would be already needed for a formal definition of the applicable CSI-faithfulness assumption.

We agree that searching for special settings that allow for polynomial decision procedures is interesting, but it is a
complicated question that needs further research. Limiting treewidth of the underlying DAG makes exact inference
polynomial for BNs, so it might work here, however, in our case some variables are unobserved. When no labels exist,
one can use polynomial ID, thus some clever limit on the labels may also work. However, in a real application there is
no guarantee that any such limits would apply.

[Meta-Review · NeurIPS 2019]

This paper studies identification of causal effects in the presence of context-specific independence (CSI), an interesting problem which is essential in some scenarios. It designs a calculus and an automated search procedure to address the problem. The works is well motivated and the method is shown to be sound.